# Meat-Free Mondays in Hospital Cafés in Aotearoa, New Zealand

**DOI:** 10.3390/nu15224797

**Published:** 2023-11-16

**Authors:** Ella Ewens, Leanne Young, Sally Mackay

**Affiliations:** 1Department of Epidemiology & Biostatistics, School of Population Health, Faculty of Medical and Health Sciences, University of Auckland, Auckland 1023, New Zealand; leanne.young@auckland.ac.nz (L.Y.);; 2National Institution for Health Innovation, School of Population Health, University of Auckland, Auckland 1023, New Zealand

**Keywords:** Meat-Free Mondays, Meatless Mondays, meat reduction, hospital, workplace food environment, sustainable healthy diets

## Abstract

Current human meat consumption levels contribute to environmental degradation and are a risk factor for non-communicable diseases. Globally, meat-reduction policy interventions are limited. Meat-Free Mondays (MFMs) is a global campaign to reduce meat consumption to improve planetary and human health. We conducted a mixed methods evaluation of MFMs at three District Health Boards (DHBs) (one not considering a MFM policy, one that had trialled MFMs and one implementing MFMs) to investigate attitudes towards MFMs and barriers and enablers to implementation. An online staff survey and eleven semi-structured interviews with food service managers, café managers and sustainability managers were conducted. Of the 194 survey participants, 51% were actively cutting back on meat, mainly for health, environmental concerns and enjoyment of plant-based dishes, and 59% were positive towards MFMs. Qualitative analysis using a general inductive approach identified four themes: (1) ‘Change and choice’ (impact on personal choice), (2) ‘Getting it right’ (product and price, food quality, health, customer retention and sales), (3) ‘Human and planetary health’ (hospitals as leaders in healthy, sustainable diets), (4) ‘Implementation success’ (communication and education). Recommendations for implementation of MFMs included seeking feedback from other DHBs, wide consultation with food service staff, cultural and dietitian food service support and providing evidence of the success of MFMs and alternatives to MFMs.

## 1. Introduction

Climate change is a serious threat to public health, potentially destroying decades of global health progress and impacting every factor necessary for human survival, such as air temperature, fresh air, drinking water, food supply and safe housing [1]. Currently, the world is not on course to achieve the emission targets necessary to limit global warming to below 2 °C compared to pre-industrial levels, following the Paris Agreement adopted in 2015 [2]. The food system is a key driver of climate change, causing pollution to soil, water, air and deforestation; reducing biodiversity; and producing greenhouse gas emissions [3]. Meat production is particularly damaging to the environment as it requires more land, water and feed to produce equivalent amounts of calories and the production of methane from ruminant animals is central to global warming [3].

Noncommunicable diseases (NCDs) are the leading cause of death in NZ and account for 89% of all worldwide deaths [4]. Of all the lifestyle risk factors for NCDs, diet is the most important [4]. The NZ diet is rich in calorie-dense, nutrient-poor foods, high in fat, salt and sugar and most New Zealanders are eating below the recommended intake of vegetables, fruits and fibre and above the recommended intake of takeaways, protein, saturated fat and red and processed meat [5]. The most recent NZ Adult Nutrition Survey indicates (2008/09) that only 32.8% of NZ adults meet the combined recommended vegetable and fruit intake of at least three servings of vegetables and at least two servings of fruit each day [5]. For fibre, NZ adults are consuming less than the daily adequate intake of 30 g and 25 g, respectively, meaning that the intakes for male and female adults were 22.1 g and 17.5 g [5].

NZ has the sixth highest meat intake per capita in the Organisation for Economic and Development (OECD) [6]. Meat is a source of B vitamins, minerals and protein; however, it is also a major source of saturated fat, and salt in the case of processed meats, in NZ diets [7]. The long-term consumption of increasing amounts of red meat, particularly processed meat, is associated with an increased risk of total mortality, cardiovascular disease, cancers such as colorectal cancer and Type 2 diabetes in both women and men [8]. Plant-based diets can reduce the relative risk of coronary artery disease, cerebrovascular disease events, Type 2 diabetes and some cancers [9,10]. Diets lower in meat are also associated with lower body mass index (BMI), waist circumference and obesity risk [11,12]. The World Health Organization (WHO), the World Cancer Research Fund, the NZ Heart Foundation and the NZ Eating and Activity Guidelines (EAGs) recommend limiting the intake of red and processed meat [13,14,15].

In 2019, the EAT-Lancet Commission recommended a largely plant-based diet, advising that profound changes must be made to the global food system to feed the world’s population of 10 billion within planetary boundaries by 2050 [16]. There is a strong scientific consensus that a dietary shift towards plant-based foods and reduced meat intake will reduce the negative environmental impact of the food system and the prevalence of deaths and health-related costs associated with NCDs [3,16,17,18,19]. However, despite this consensus, there has been little policy action on meat reduction in NZ or globally and meat intake remains high [19,20].

Meatless Mondays (MMs) is a global meat reduction campaign, originating in the US, which encourages people to forgo meat one day a week, reduce consumption by 15% and eat more plant-based foods to improve health and reduce the environmental burden connected with meat production [21]. MMs and MFMs (a variation of MMs) have been implemented in restaurants, schools, hospitals and workplaces in at least 45 countries worldwide. At institutions where MMs have been implemented, menus are made up of a majority of plant-based and include some meat, whereas MFM menus are completely meat-free. There are few studies on the effectiveness of MMs or MFMs; however, research indicates that MMs in school settings result in significantly lower food-related greenhouse gas emissions, less water and land use, and are comparable in cost and nutrition [22]. In hospitals, where MMs have been implemented, evaluations indicate evidence of behaviour change beyond Mondays, such as eating more fruit and vegetables, reducing meat intake, and eating more plant-based options at home or when dining out. Additionally, MMs resulted in more plant-based meals being purchased, increased awareness of MFMs and increased awareness of public health messages associated with the impact of meat on health and the environment [23].

Te Whatu Ora, Health New Zealand, is the largest employer in NZ and a substantial emitter of greenhouse gases [24,25]. In 2020, the Nelson Marlborough District Health Board (DHB1) was the first workplace in NZ to launch a MFM policy at their staff and guest cafés as an extension of the existing National Healthy Food and Drink Policy (NHFDP). They offer a completely meat-free menu on Monday, a Fish Friday and also removed all processed meats from café menus. This study aimed to identify the enablers and barriers of MFM policies at three District Health Boards (DHBs) to offer policy recommendations for the future to facilitate implementation.

## 2. Materials and Methods

### 2.1. Locations

Three DHBs were selected (of the total 20 DHBs in NZ) at different stages of contemplation or implementation of MFMs. Nelson Marlborough DHB (DHB1) had been enacting the policy for approximately two years, Northland DHB (DHB2) had run an eight-week trial of MFMs in their staff café and Auckland DHB (DHB3) was not considering implementing an MFM policy. The reason for the heterogeneity of the research group in location and level of participation in MFMs was to gain a range of data and experiences from DHBs at different stages of implementation or not of the MFM policy.

### 2.2. Online Staff Surveys 

The 12-question survey was formulated with feedback from the research team of three nutrition experts, three lay people and one DHB manager. (Appendix A Meat-Free Mondays Survey). Online staff surveys were conducted at DHB3 and DHB1, but not DHB2 as they had previously informally surveyed staff about MFMs. DHB2 was included in the interviews. The survey designed on Qualtrics XM 2022 contained three key sections: (1) demographics, (2) eating habits and ‘cutting back on meat’ behaviours, (3) attitudes towards MFMs, enablers (e.g., health, environmental, cost, taste), perceived benefits, barriers (e.g., taste, health, familiarity and special diets) and impact on future behaviours. DHB staff were invited via email and staff intranet to complete the survey anonymously. Participation was available between 20 February and 13 April 2022 and was voluntary. Participants could only be identified if they opted to receive a summary of the study results via email. The aim was to gain a proportionally representative sample size of 50 staff, with Māori and Pacific responses equal to that of the population studied; however, this was not reached in the designated time period, and therefore, the survey remained open for an additional week until the 20 April 2022, during which a reminder email was sent to all staff. 

### 2.3. Survey Statistical Analysis

Proportions of participants were calculated for sociodemographic groups: DHB, age group, ethnic group and job role. However, due to very small numbers in some ethnic groups, ethnicity was not used in the statistical analysis. Job roles were regrouped into ‘Doctor’, ‘Nurse and midwife’, ‘Allied health or other health professional’, ‘Admin and support staff’, ‘Academic staff and Management’ and ‘Other’ to allow for small numbers of participants. Statistical analysis could not be conducted where two job types were selected; therefore, minority prioritisation was used. Survey data results were analysed using χ^2^ Chi-squared testing *p* < 0.05. All analyses were conducted using IBM SPSS statistical software 29 2022.

### 2.4. Semi-Structured Interviews

Participants from each of the three DHBs were selected from three key job roles: (1) food service managers, (2) café managers and (3) sustainability managers. At DHB1, a public health advocate who was closely involved in initiating and implementing the policy was invited to participate. A national food service manager from a large catering company supplying 12 DHBs was also invited to participate due to their involvement in MFM policy implementation. Purposive sampling was used to select interviewees based on relevant knowledge, experience and availability. Participants were identified by a liaison person at each DHB, who provided contact details for each participant. Participants were contacted directly via email or phone to invite them to be interviewed. At two of the DHBs, the café manager and food service manager were both present at the interview and interviewed sequentially. This interviewing style was completed at the request of the interviewees as it was most convenient and provided a culturally safe, comfortable and amenable interview environment for the participants. The duration of the interviews ranged from 30–60 min. Interviews were recorded and transcribed using Zoom.

### 2.5. Interview and Open-Ended Survey Questions Analysis

A general inductive approach with QSR NVivo 12 was used to analyse the interview and online survey text. This technique was appropriate to condense widespread and mixed raw data into a succinct summary format and identify clear connections between the research objectives related to the perceived barriers and enablers of MFMs in DHBs. This general inductive approach was selected as there were clear research objectives and the approach is flexible and efficient, and can also help to link themes, ideas and meanings and help predict outcomes or create new theories [26,27,28]. Transcriptions from Zoom were cleaned (transcription errors removed) and imported into NVivo, where they were read multiple times until the content became familiar. Codes, aligned with the study aims and objectives, were derived using close reading, these were then confirmed by a second researcher through discussion and review. Codes were then edited and refined, and the number of codes was reduced from 70 to 36. Overlapping codes, redundant codes and repetitive codes were removed, and the remaining codes were grouped to create themes. A brief description was written that explained each theme and example quotes were selected to convey the key elements of each theme, which was confirmed by a second researcher. 

### 2.6. Ethics Approval

The Auckland Health Research Ethics Committee approved this study on 12 August 2020 for three years (reference number AH2519). Approval to conduct the research in each DHB was obtained in writing from each DHB liaison dietitian following ethics approval.

## 3. Results

Results of the surveys at DHB1 and DHB3 are presented first, followed by the interview results at DHB1, DHB2 and DHB3.

### 3.1. Sampling

The online survey received 194 responses, 105 at DHB3 (45.9%) and 89 at DHB1 (54.1%). Table 1 shows most participants were female (76.3%), Pākehā (72.7%), allied health or other health professionals (28.9%). The survey responses across different job types resembled the job demographics of the staff at the two DHBs involved [23,27]. 

### 3.2. Eating Behaviours

Most respondents were meat eaters (63.4%), 18.6% were flexitarian, 7.2% were pescatarian, 6.7% were vegetarian and 4.1% were vegan across both DHBs surveyed (Table 2). Of those who reported eating meat (e.g., red meat, poultry), 50.5% responded that they were actively cutting back on meat intake. Approximately one and a half times as many respondents were vegan at DHB3 (12.4%) compared with DHB1 (8.9%). There were approximately twice as many flexitarians at DHB1 (25.8%) than at DHB3 (12.4%). In the 44 years and under age group, there were more vegans and vegetarians than in the 45 years and older age group. For participants who responded they were eating meat (i.e., meat eaters and flexitarians) (N = 159), most reported eating meat 5–6 times per week (37.1%) followed by 3–4 times per week (25.2%) across both DHBs, with 17.9% eating meat seven times per week. No statistical difference was found between the two DHBs in the number of times meat was consumed per week χ^2^(4, N = 194) 2.37, *p* = 0.669. A higher percentage of staff from DHB3 (55.8%) were cutting back on meat compared with DHB1 (44.9%). However, this difference was not statistically significant χ^2^(1, N = 194) 2.25, *p* = 0.134. 

### 3.3. Enablers and Barriers to Cutting Back on Meat

Table 3 shows the enablers and barriers to cutting back on meat at DHB1 and DHB3. Of those cutting back on meat, the main reasons selected were health (32.0%), environmental concerns (30.4%), enjoying plant-based dishes (30.4%), animal welfare (19.6%) and saving money (14.9%). At DHB3, ‘environmental concerns’ was the top motivator for cutting back on meat (35.2%) with a lower proportion at DHB1 (24.7%), although not significantly different χ^2^(1, N = 194) 2.519, *p* = 0.113. The main reasons reported for not cutting back on meat were thinking that it is part of a healthy diet (30.1%), taste (26.3%) and familiarity (16%). More respondents from DHB1 selected ‘attitudes of friends, whānau and or family’ as a barrier to cutting back on meat (10.1%) compared with DHB3 (2.9%). This difference was found to be statistically significant χ^2^(1, N = 194) 4.368, *p* = 0.037. Across both DHBs, 9.8% selected ‘Other’, and reasons included medical, iron, protein, lack of meat alternatives, meat as a natural food and living on a farm.

### 3.4. Awareness and Support for MFMs

Overall, awareness of the global MFM campaign was high (65.5% across both DHBs), although significantly higher in DHB1 (79.8%) compared with DHB3 (53.3%) χ^2^(1, N = 194) 14.90, *p* = 0.001 (Table 4). The awareness of MFMs in NZ hospitals was lower than awareness of the global campaign, with 76.4% aware at DHB1 and 15.2% aware at DHB3, χ^2^(1, N = 194) 73.41, *p* = 0.001. Overall, there was support for an MFM policy (or potential policy) in both DHB1 (55.1%) and DHB3 (61.91%). The level of positive/negative feelings reported towards MFM policies was not statistically significant between DHBs, χ^2^(4, N = 194) 3.69, *p* = 0.450.

Overall, 58.8% were very positive or positive towards an MFM policy (or potential policy), 10.3% were neutral and 31.0% were negative or very negative. There were more positive views for MFMs amongst vegans and vegetarians compared with flexitarians, pescatarians and meat eaters, and no negative responses to MFMs amongst vegans and vegetarians combined.

### 3.5. Staff Surveys and Interviews 

Eleven semi-structured interviews were held with food service managers, café managers and sustainability managers (referred to as managers) from all three DHBs: DHB1, DHB2 and DHB3. All participants invited agreed to participate. Table 5 shows the participants and their DHB locations. Two interviews were conducted with two participants present; therefore, the total number of interviews was nine. The codes and themes from both the written responses to the online staff survey (customers) and the manager interviews were similar, therefore they are reported together within each theme and supporting quotes. Four key themes were identified. Table 6 shows each theme and codes within each theme.

#### 3.5.1. Change and Choice

-Overcoming resistance to change and promoting choice with plant-based foods.

This theme explored the importance of personal choice in eating habits and hospital café menu options. Limiting personal choice was found to be a commonly discussed factor in the negative feedback about MFMs from both customers and managers. The responses implied that limiting personal choice on hospital café menus was unethical. This theme also includes the sub-theme, resistance to change—both from an individual level, regarding a change to individual life-long eating habits/food choices, but also resistance to policy change at the organisational level. This theme includes the challenges people experience when trying new foods and the challenges of behaviour change, including sociocultural aspects, such as family or religious traditions, related to MFMs. Promoting choice in hospital café menus was seen as an important component for the success of an MFM policy. 


*It can be hard to change the habits of a lifetime.*



*So, for short term, I can see it will be a little bit of a sales drop. But after the whole environment, the community... people accept it, they pass (it on by) verbal communication or pass it on one by one. This will affect more people, when the majority accept it, I think there won’t be any problem anymore.*



*People/Organisations shouldn’t be imposing their ‘righteous’ beliefs on others. If people want to eat meat, then they should have that choice, just like if someone wants to be meat-free then that should also be a choice on the menu.*



*Find another way—I’m over not having choices.*


#### 3.5.2. Getting it Right—Product and Price 

-Delivering delicious, affordable and nutritionally balanced food for café success.

This theme explored the importance of getting the food products and menus right in terms of providing tasty, nutritionally balanced and affordable meals at DHB staff cafés. Objections to MFM meals included concerns about iron, protein and special diets such as low-carb diets. Some customers emphasised the need for nutritionally balanced MFM meals, particularly regarding protein. Hospital staff working long hours expressed concerns about satiety and stated that meat-inclusive meals may be more satiating and comforting. This theme also considered the potential impacts of MFMs on customer retention in the cafés, especially when considering nearby competitor food suppliers. 


*If the current vegetarian options are anything to go by, then I wouldn’t be happy with Meatless Monday. If they had better options, I would be all on board.*



*No, I don’t believe customer numbers have (reduced), I think in a way we have managed to bypass that and find really good alternatives in those places that we operate (MFMs) in.*



*…appearance and taste, obviously, tasty, good vegetarian options, you can’t just have stodgy macaroni cheese.*


#### 3.5.3. Human and Planetary Health 

-Driving education, awareness and advocacy amongst key stakeholders.

This theme explores the purpose of a MFM policy and the co-benefits for human and planetary health from MFMs in NZ hospital cafés. It also includes how staff and customers believed that hospitals are well-positioned to be leaders in promoting and providing healthy and sustainable diets.


*This is a positive move that aligns the hospital with a huge body of scientific data that promotes reducing meat consumption and increasing vegetable consumption. We should be leaders in health, not only in treating sickness.*



*It’s important for hospitals and health services to lead by example with regards healthy food choices.*



*You know it doesn’t make any sense to be selling food that causes problems that our hospitals are trying to fix up in the first place. And I think you know five or 10 years down the track, we will realise that, eating red meat is not ideal, not ideal for us, it’s not ideal for planet, it’s not ideal for the animals.*


#### 3.5.4. Implementation Success 

-Implementing a successful Meat-Free Monday policy and achieving optimal food service staff wellbeing.

This theme includes the necessary steps involved in implementing a MFM policy, including consultation, communication, education and evaluation to achieve success and acceptance from customers and maintain optimal wellbeing for food service staff. 


*I feel like it would be more productive to increase the amount of vegetarian/vegan options throughout the week. I look forward to meat-free Mondays because it is the only time there is some actual variety, every other day only has one vegetarian option and it’s usually the same sort of thing every time and often it isn’t vegan. This is particularly frustrating as someone with allergies.*



*Training with the staff, you know, don’t just tell them it’s going to happen. Support them behind it, give them a book, give them some answers for when people have them on, because throwing things at people’s faces is just too much.*



*It’s really important that staff are fully consulted with and that there’s a really good communications package… there’s lots of notice, it’s repeated regularly to the staff…the worst thing that can happen is that staff working on day one or week two or whenever it is turn up …desperately, seeking a meat pie…*


The interviews and surveys revealed the importance of good communication and consultation with staff for the success of a MFM policy. This was considered a crucial step to give everyone optimal notice of the change and a chance to feed back, to avoid surprises. Education, training and notification were important steps to gain buy-in from food service and café staff and for them to take ownership of the MFM policy and successfully implement it. In addition, providing hands-on training (including testing the food options), support such as recipe ideas and nutritional information, as well as information on the benefits of the policy were considered important during the rollout of the policy.

## 4. Discussion

This study showed that DHB staff had a strong interest in and general awareness of MFMs. Half of staff (51%) reported to be actively cutting back on meat and most supported MFMs, with key motivators identified as health, environment concerns and enjoyment of plant-based meals. These meat reduction results were higher than a 2019 NZ consumer poll (N = 1000), which showed 31% of people were cutting back on meat [29]. The healthcare workplace setting is unique as staff have a high level of education and a focus on behaviours that support personal health; this could explain the finding of health as a key motivator for meat reduction [30]. However, just below one-fifth of respondents identified as flexitarian, compared to a recent NZ consumer poll which indicated a third of New Zealanders are flexitarian, which may suggest an inconsistency or lack of awareness or understanding of the term ‘flexitarianism’ in the 2019 poll and in the present study [29].

The results of this study are consistent with other research into the enablers of meat reduction, where health and environment are cited as key drivers [31,32,33]. However, most research cites health as the most important motivator [32]. Interestingly, amongst DHB3 staff, environmental concerns were the most prevalent factor for reducing meat, albeit only slightly higher than health, which perhaps indicates an increasing awareness of the impact of food choices on the environment at this location. 

Other studies have found ‘trying new foods’ to be an important motivator, similar to the motivator found in this study of ‘enjoying plant-based foods.’ An unexpected finding was the perception that as part of MFMs, New Zealanders could broaden their food horizons by trying new plant-based foods and this behaviour could positively impact behaviour on other days of the week, e.g., cooking at home and eating out. This result is encouraging in NZ, where meat eating levels are high compared to other high-income countries (sixth in the OECD) and high levels of meat eating are considered status quo—a constant in our farming history and economy and considered part of our identity [6].

A proportion of respondents did not feel that reducing meat would positively impact the environment. Some felt other sustainable actions would have more benefits, such as packaging, recycling, reducing food transportation and regenerative agriculture. This sentiment was echoed in the interviews and is aligned with other research in this area. A systematic review of studies from Australia, the Netherlands, Belgium, Switzerland, and the United Kingdom, found that less than a quarter of respondents were aware of the environmental impact of meat [34]. Another study showed that of those who consider the environmental aspects of food, other aspects such as food production and distribution such as transport, deforestation, pollution, and packaging were considered of greater importance than the food choice itself [35].

Awareness of the global MFM initiative was high and similar in both DHBs surveyed, which is a testament to the marketing efforts of overseas MFM groups, such as in the UK and US, to raise awareness of the campaign [21]. As expected, awareness of MFMs in NZ hospitals was higher in DHB1, where the policy had been running for two years. However, awareness of MFMs itself does not necessarily indicate an understanding of the reasons behind MFMs, nor does it indicate that MFM has changed eating behaviours on Mondays (e.g., if staff are not eating at the cafés involved) or on other days of the week. Previous research in the US has shown that MFMs can increase awareness of health and environment-focused messages about MFMs, meat intake and production (perceived message effectiveness) [23,36]. MFMs have also been shown to positively impact eating behaviours on other days of the week, e.g., eating more fruits and vegetables and cooking more plant-based meals at home [23]. As this study did not evaluate these behavioural effects, further research in this area is required.

While some objected to MFMs, almost two-thirds of the staff at both DHBs surveyed supported a MFM policy in its current format, with the complete omission of meat on Mondays. Interestingly, there was no significant difference in how positive respondents were towards MFMs at DHB1 where MFMs were operating, compared to DHB3, where they were not operating. This indicates that while there was some customer resistance, overall, the MFM policy has not substantially negatively impacted attitudes towards MFMs at DHB1.

Some participants felt that MFMs limited personal choice, consistent with the Norwegian Armed Forces’ research on MFM challenges due to interference with individual choices [37]. New Zealanders, especially hospital workers, faced strict COVID-19 orders (mask-wearing, social distancing, testing, isolating, lockdowns), requiring motivation to change behaviour and beliefs about policy benefits [36]. Hospital staff, already impacted by COVID-19 restrictions, were sensitive to further choice limitations, possibly affecting study responses at this time. Paternalistic public health policies (e.g., COVID-19 lockdowns, MFMs) raise debates on balancing public health and individual liberties [38,39,40,41]. Some respondents supported educational approaches for healthy, sustainable foods but were less supportive of MFMs impacting personal choice.

Hospital staff expressed dissatisfaction with perceived rights violations, defending meat consumption and opposing management’s beliefs. Staff’s preference for meat may be due to personal taste rather than a fundamental right. Doctors and nurses, with limited time, often rely on hospital cafés and also benefit from free meals. Despite restrictions, they could still eat meat by bringing food or using other outlets.

MFMs removed meat from hospital cafés for only one meal, once a week (4.8% of weekly meals). Considering individual opposition, it is vital to weigh policy impact with overall quality of life in enduring public health debates. The importance of hospital cafés providing tasty, balanced and affordable meals was identified in the current study. Objections to MFM meals included concerns about special diets (such as low carb or Keto) and nutritional considerations like protein and iron content. Some staff expressed that meat-inclusive meals were more satiating when working long hours in surgery. However, plant-based meals can also provide adequate protein through sources like soy products, legumes, whole grains, nuts, and seeds, which offer satiety through their high fibre, protein and water content. Protein deficiency is rare among New Zealanders [7].

Food service managers faced challenges in providing varied, seasonal and affordable meals, with cost implications being a sensitive topic due to the rising cost of ingredients and decreased sales during the COVID-19 pandemic. There was no consensus on whether MFMs would save money, as some perceived cost benefits using legumes over meat, while others worried about losing customers to competitor cafés when not serving meat options on Mondays. Overall, perceptions in this study differed from studies in high-income countries that show lower food costs with flexitarian meals [40,41,42]. One analysis indicated that healthy and sustainable diets, e.g., flexitarian, pescatarian, vegetarian and vegan could result in 22–34% lower food costs in upper–middle income to high-income countries such as the UK, Europe, Australia and NZ. It was an unexpected finding in this study, that hospital cafés were using expensive ready-made vegetarian alternatives such as vegan sausage rolls and plant-based burgers due to perceived consumer demand, with limited discussion amongst managers in the surveys about the use of cheaper options like legumes.

Participants acknowledged the environmental benefits of MFMs, such as reduced methane emissions and biodiversity loss, with sustainability managers being more confident and knowledgeable about these benefits compared to food service or café managers. Some food service managers were unsure about the health benefits of reducing red or processed meat intake, indicating a need for further education. Overall, most participants believed that hospitals had the opportunity to lead in health and sustainability by raising awareness amongst staff and visitors about healthy and sustainable eating options. If MFMs were implemented nationwide, it would make a strong statement about health and sustainability; however, there may be initial backlash from the farming industry and other groups.

Successful implementation of MFM policies in NZ requires thorough consultation with DHB staff, management, food service, dietitians and cultural advisors. Engaging food service and café staff is crucial for the success of MFMs, as seen in previous research [37]. Clear communication, training and information sharing are essential to address concerns and gain buy-in from staff. Educational resources should emphasise the benefits of MFMs for human and planetary health, providing evidence-based explanations for the policy. Resources for café staff and customers could include information on the benefits of meat reduction, recipes, and frequently asked questions.

This study identified the misconception that MFMs aim to convert people to vegetarianism, which may hinder the success of MFMs. Clear information on the aims of MFMs is necessary. Evaluation of MFMs should include metrics such as participation, engagement and acceptance of MFMs, customer numbers in cafés, sales figures and awareness of associated public health messages. Resistance to changing eating habits, particularly due to sociocultural factors, was evident in this study. While most participants supported MFMs, alternatives were suggested, such as using positive language or increasing the ratio of plant-based options throughout the week. The ‘Meat-less’ approach used in MMs in the US, with a 3:1 ratio of plant-based to meat options, was favoured by several respondents. A mandated MFM with complete omission of meat on one day challenges personal choices and aims to change strong social norms around meat eating which make reducing meat intake challenging for individuals. However, broader concurrent education on the impact of meat consumption on health and the environment is needed, with the responsibility falling on public health practitioners and the education system. 

For the success of MFMs, it is important to consider how supportive the wider food provision policy environment in NZ is of sustainable eating and meat reduction. The greater food environment such as the food outlets in the hospital vicinity can be considered broadly unsupportive of MFMs and meat reduction in NZ. While the Eating and Activity guidelines for NZ Adults and the NHFDP in hospitals mention sustainability, it is a very small component of these guidelines and there are no strong policies or implementation to support it [15].

### 4.1. Strengths and Weaknesses

A strength of this study is the mixed methods approach, as qualitative data offer detailed, contextualised insights and quantitative data offer externally valid, generalisable insights. Furthermore, a mixed methods approach can help to offset weaknesses inherent to one design by using both. Another strength is that data were collected from both consumers who have first-hand experience with hospital café food, and from a range of managers from different sectors at varying levels of contemplation and implementation of MFMs. All those invited to take part in the interviews participated and many respondents provided lengthy comments to the open-ended questions in the online survey, perhaps indicating strong opinions were held on the topic of MFMs.

In this study, a majority of DHB staff surveyed supported an MFM policy; however, only three DHBs out of 20 were evaluated and large sociocultural differences exist across NZ DHB regions. Considering the total population at each DHB, online survey responses were low. Email surveys often obtain a low response, and survey respondents tend to be less busy, have a higher level of agreeableness, higher ‘extraversion’ (personality trait measured by psychometric testing) and want their opinions (either positive or negative) to be heard [43]. Additionally, Likert scale responses are susceptible to central tendency bias [44]. However, the results in this survey were not biased towards the centre, but were polarised towards the extremities, which also indicates people held strong views towards MFMs. Most survey respondents were female (76.3%) and allied health professionals, although numbers were representative of employees at DHBs involved [24,45].

### 4.2. Further Research

Further evaluation of the support of other NZ DHB regions for MFMs should be conducted before a national MFM policy is rolled out. As some staff cafés are open to the public, assessment of public support should also be considered. The research focused on food provided to DHB staff and café visitors, not patients staying in hospitals. Patient food presents another opportunity to reduce greenhouse gas emissions, raise awareness of the environmental and health impacts of meat production and save costs, and therefore, future policies should tackle this area. More research is also required to evaluate and quantify the environmental impact of MFMs in the NZ setting. Utilising the NZ lifecycle database of foods, the environmental impact (kgCO_2_e/kg) of MFM meals compared with meat-inclusive meals could be calculated [46]. Evaluation of water and land usage of foods would also be useful for an NZ setting, as this would differ from calculations completed abroad. Investigation of supply (and customer) costs of meat-free menus versus other days of the week would also be beneficial. Further research could also include interviews with public health dietitians and measurement of the nutritional profile of meals provided as part of MFMs, this may aid in alleviating concerns around the nutritional balance. Further research is also necessary to investigate how MFM policies can change eating behaviours relating to reducing population meat intake as current research in this area is limited.

### 4.3. Policy Recommendations

For successful food policy change in hospital cafés, consultation, education and support is crucial. A mandatory national MFM policy should be considered if there is majority support across NZ, with the alternative of a meat-less approach considered. Measures to promote plant-based options and engage key stakeholders are recommended. Consultation should involve key stakeholders, staff, public health dietitians and cultural advisors. Information sharing, training and evidence-based education to support the environmental and health benefits of the campaign are important for management and hospital staff. Support for food service staff should include training, recipes, marketing materials and evaluation tools. General recommendations include offering appealing, tasty and nutritionally balanced options, menu variety, increasing plant-based options and removing processed meats. Future recommendations include evaluation of the policy, accountability processes, information-sharing forums, widespread promotion and plans for how to address policy breaches.

## 5. Conclusions

This research examined MFM policies at three DHBs in NZ at different stages of MFM implementation and identified enablers and barriers at organisational and individual levels. Awareness of MFMs was high, with strong support from both staff and managers, despite some criticism. Resistance of staff to change behaviour, concerns around special diets and retaining personal choice were identified as key barriers. Over half of the surveyed staff were reducing their meat intake, driven by health and environmental concerns. The qualitative analysis uncovered key themes, including the importance of food quality and appearance for business success and customer retention, communication, education and implementation success, and the overarching significance of hospitals taking the lead in human and planetary health and fostering healthy, sustainable diets.

Policy recommendations for successful implementation of MFMs include wide consultation with food service, management, hospital staff and cultural advisors; providing clear aims and alternatives; considering the wider food environment (including NZ food policy and food outlets in the hospital area); and providing full training and support for the food service, including dietitian support. Communication, education and providing evidence of the benefits of MFMs are crucial factors for a successful MFM policy roll-out. Evaluation of the policy’s environmental impact and costs will be valuable in the future as the health system remains a high emitter of greenhouse gas emissions. MFM policies can drive awareness of public health messages about the impact of meat consumption, and despite barriers, comprehensive implementation is likely to make MFMs successful in healthcare settings.

## Figures and Tables

**Table 1 nutrients-15-04797-t001:** Demographics of survey respondents by DHB and overall.

	DHB 3		DHB 1		Total	
**Responses**	n	%	n	%	n	%
	105	51.1	89	45.9	194	100
**Gender**						
Female	85	81	63	70.8	148	76.3
Male	18	17.1	24	27	42	21.6
Gender diverse	0	0	0	0	0	0
Prefer not to say	2	1.9	2	2.2	4	2.1
**Age**						
18–24 years	6	5.7	6	6.7	12	6.2
25–34 years	23	21.9	29	32.6	52	26.8
35–44 years	26	24.8	16	18.0	42	21.6
45–54 years	26	24.8	18	20.2	44	22.7
55–64 years	20	19.0	16	18.0	36	18.6
65 years and over	4	3.8	4	4.5	8	4.1
**Ethnic group ***						
NZ European/Pākehā	70	66.7	73	82	141	72.7
Māori	5	4.8	6	6.7	13	6.7
Samoan	2	1.9	0	0	2	1
Chinese	6	5.7	1	1	7	3.6
Indian	9	8.6	0	0	9	4.6
Other	20	19.0	13	14.6	33	17
**Job type**						
Medical doctor	13	12.4	10	11.2	23	11.9
Nurse and midwife	21	20.0	20	22.5	41	21.1
Allied health or other health professional	34	32.4	22	24.7	56	28.9
Admin and support staff	14	13.3	20	22.5	34	17.5
Academic staff and Management	13	12.4	10	11.2	23	11.9
Other	10	9.5	7	7.9	17	8.8

Note. * Ethnic group percentages do not add up to 100% due to the option of selecting more than one ethnic group.

**Table 2 nutrients-15-04797-t002:** Meat eating behaviours by District Health Board.

	DHB3		DHB1		Total	
	n	%	n	%	n	%
**Eating patterns ***						
Meat eater	71	67.6	52	58.4	123	63.4
Flexitarian	13	12.4	23	25.8	36	18.6
Pescatarian	8	7.6	6	6.7	14	7.2
Vegetarian	7	6.7	6	6.7	13	6.7
Vegan **	6	5.7	2	2.2	8	4.1
**Times per week eating meat**						
<1	4	4.8	4	5.3	8	5
1–2	10	11.9	14	18.7	24	15.1
3–4	23	27.4	17	22.7	40	25.2
5–6	30	35.7	29	38.7	59	37.1
7+	17	20.2	11	14.7	28	17.6
*p* = 0.699						
**Are you actively cutting back on meat?**						
Yes	58	55.8	40	44.9	98	50.8
No	46	44.2	49	55.1	95	49.2

*p* = 0.134. * Flexitarian = I am primarily vegetarian, but occasionally eat fish or meat, including chicken—less than one portion of red meat/week. Pescatarian = I eat fish, but no other meat types. Vegetarian = I do not eat meat or fish, I may or may not eat dairy and/or eggs. Vegan = I do not eat any animal products. ** No statistical testing was completed for Eating patterns.

**Table 3 nutrients-15-04797-t003:** Enablers and barriers from Surveys for cutting back on meat at DHB1 and DHB3 (n = 194).

Enablers	Barriers
Environmental concerns	I like the taste of meat
Health	I think meat is part of a healthy diet
I enjoy plant-based dishes	I am familiar with eating meat. It’s what I am used to
Animal Welfare	Other
Saving Money	I don’t believe reducing meat intake will have any impact on the environment
Other	Attitudes of friends, whānau and/or family
	It’s not culturally appropriate to exclude meat

**Table 4 nutrients-15-04797-t004:** Awareness and support of global and New Zealand Meat-Free Mondays by District Health Board.

	DHB3		DHB1		Total	
	n	%	n	%	n	%
**Awareness of the global Meatless Monday campaign**						
Yes	56	53.3	71	79.8	127	65.5
No	49	46.7	18	20.2	67	34.5
*p* = 0.001						
**Awareness of Meat-Free Mondays in NZ hospitals**						
Yes	16	15.2	68	76.4	84	43.3
No	89	84.9	21	23.6	110	56.7
*p* = 0.001						
**How do you feel about a Meat-Free Monday policy or potential policy at your hospital?**						
Very positive	50	47.6	34	38.2	84	43.3
Positive	15	14.3	15	16.9	30	15.5
Neutral	11	10.5	9	10.1	20	10.3
Negative	12	11.4	18	20.2	30	15.5
Very negative	17	16.2	13	14.6	30	15.5
*p* = 0.450						

**Table 5 nutrients-15-04797-t005:** Interview participants’ job roles and DHBs.

	DHB	Job Role
1	DHB3	Sustainability manager
2	DHB3	Franchisee café manager
3	DHB3	National food service manager (Franchisee)
4	DHB2	Food service manager
5	DHB2	Sustainability manager
6	DHB2	Café manager
7	DHB1	Café manager
8	DHB1	Food service manager
9	DHB1	Sustainability manager
10	DHB1	Public health professional
11	DHB1	Food service manager

**Table 6 nutrients-15-04797-t006:** Themes from surveys and interviews with associated codes.

**1. CHANGE AND CHOICE**
Changing behaviour and social norms
Cultural, generational, social aspects
Customer backlash
Politics, dictatorship, extremism
Silent majority vs. vocal minority
Eating habits, preferences, ‘medical’ requirements
Meat eating as ‘normal’, habitual, comforting or rewarding
Personal choice, options, variety available
Plant-based foods as trendy or in demand
Resistance and time to change
Stigma of vegetarianism, veganism
Trying new things
**2. GETTING IT RIGHT—PRODUCT AND PRICE**
Cost benefits
Costs and financial incentives
Customer retention and sales
Food environment, competition
Food taste, quality and appearance, macronutrients
**3. HUMAN AND PLANETARY HEALTH**
Animal welfare, ethics, corporate responsibility
Co-benefits
Environmental benefits
Global impacts, necessity to change the way we eat
Health benefits
Hospitals as leaders in health
Lack of confidence in MFM, pessimism
Lack of understanding of health or sustainability benefits
Perception about MFM purpose
Positive feedback
**4. IMPLEMENTATION SUCCESS**
Advocacy
Buy-in and engaging stakeholders
Consultation
Education and awareness
Meat-free vs. meatless alternatives to MFM
Media, communications, marketing
Notification
Staff wellbeing
Timing of policy, competing priorities, COVID-19
Training, resources, objection handling
Trials, follow-up, evaluation, collaboration, refinement

## Data Availability

The data presented in this study are available on request from the corresponding author. The data are not publicly available due to the ethical guidelines that participants in this study consented to, which state that all future use of the data collected is controlled in accordance with the New Zealand Privacy Act, 1993.

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
