# Peer review of "Meat-Free Mondays in Hospital Cafés in Aotearoa, New Zealand"

_nutrients, 2023, doi:10.3390/nu15224797_

Round 1
Reviewer 1 Report
Comments and Suggestions for Authors
Thank you for submitting the manuscript "Meat-Free Mondays in Hospital Cafés in Aotearoa, New Zealand" to Foods. MFM policy has been included across the planet, but little is said about the impact of this inclusion and therefore, the manuscript is interesting from this point of view. Overall, I have some suggestions to make to improve the quality of the manuscript.
- adapt reference citations to the journal's standard.
- Line#47: first time the abbreviation appears and therefore needs to include the definition.
- Line#91: it seems to me that the intention was to include boards at different stages of implementation or not of the MFM policy. A justification for the heterogeneity of the researched group must be included.
- Line#97: include numbers of people who were used to formulate the survey.
-Line#98: it is not clear whether DHB2 data would be included in the manuscript or not.
- Line#102: consider including the survey as a supplementary file.
- Line#109: consider including the extra time that the questionnaire was open. Is it already described in full time on line#104?
- Line#243: is the location of this statement correct? I didn't understand his connection with the previous text.
- Line#324: I believe it is good to make it clear that the lack of understanding of the concept of "flexitarianism" occurred in the work cited and not in the present work. That's what you meant, right?
Author Response
Ella Ewens
University of Auckland,
Department of Epidemiology & Biostatistics, School of Population Health, Faculty of Medical and Health Sciences, University of Auckland, Auckland 1023, New Zealand
Ella.ewens@gmail.com
26th October 2023
Subject: Minor Revisions of Manuscript following Peer Review - "Meat Free Mondays in Hospital Cafes in Aotearoa, New Zealand"
Dear Tracy assistant editor, editors and reviewers,
Thank you for your time and consideration in the review of our manuscript. Your care in reviewing the manuscript and interest in the topic is much appreciated.
I am writing to respond to the minor revisions of the manuscript titled "Meat Free Mondays in Hospital Cafes in Aotearoa, New Zealand" from the three peer reviewers, for publication in Nutrients MDPI.
Below are the reviewer comments, and I have outlined how these comments and revisions have been addressed.
|
Reviewer Comment |
Response and actions made |
|
Reviewer 1 |
|
|
Adapt reference citations to the journal's standard. |
According to Nutrients website referencing should be in square brackets. This has now been updated. The reference list includes the full title. |
|
Line#47: first time the abbreviation appears and therefore needs to include the definition. |
I have updated OECD to the full definition in the first mention on line #47. Organisation for Economic and Development (OECD). |
|
Line#91: it seems to me that the intention was to include boards at different stages of implementation or not of the MFM policy. A justification for the heterogeneity of the researched group must be included. |
Yes, the intention was to include district health boards at different levels. I have updated the manuscript to make it clearer and to include justification of the heterogeneity of the sample. The reason for the heterogeneity of the research group in location and level of participation in MFMs, was to gain a range of data and experiences from DHBs at different stages of implementation or not of the MFM policy.
|
|
Line#97: include numbers of people who were used to formulate the survey. |
I have updated the manuscript to note the numbers. The 12-question survey was formulated with feedback from the research team of three nutrition experts, three lay people and one DHB manager.
|
|
Line#98: it is not clear whether DHB2 data would be included in the manuscript or not |
I have updated the manuscript to make it clearer. DHB2 was included in the interviews. |
|
- Line#102: consider including the survey as a supplementary file. |
I have added this as a supplementary file. I have referred to it in the text, as per Nutrients guidelines. (Document S1 Meat-Free Mondays Survey). |
|
- Line#109: consider including the extra time that the questionnaire was open. Is it already described in full time on line #104? |
I have updated the manuscript to improve clarity. Therefore, the survey remained open for an additional week until the 20th April 2022, during which a reminder email was sent to all staff.
|
|
Line#243: is the location of this statement correct? I didn't understand his connection with the previous text. |
“It can be hard to change the habits of a lifetime.” This quote relates to difficulty in changing eating habits, I think that it is mentioned in the text above. I have added some more words to make it clearer. I have updated the manuscript to improve clarity. This theme also includes the sub-theme, resistance to change – both from an individual level, regarding a change to individual life-long eating habits/food choices, but also resistance to policy change at the organisational level. |
|
- Line#324: I believe it is good to make it clear that the lack of understanding of the concept of "flexitarianism" occurred in the work cited and not in the present work. That's what you meant, right? |
I was meaning the lack of understanding occurred both in the present study and in the poll. I have updated the manuscript to make it clearer. However, just below one-fifth of respondents identified as flexitarian, compared to a recent NZ consumer poll which indicated a third of New Zealanders were flexitarian, which may suggest an inconsistency or lack of awareness or understanding of the term ‘flexitarianism’ in the 2019 poll and in the present study. |
Thankyou again for your thorough review of the manuscript and for your helpful and insightful comments.
I appreciate the opportunity to contribute to the body of evidence on sustainable dietary practices.
Sincerely,
Ella Ewens
MPH(Hons), BOptom(Hons), PGDipHsc, PGDipBus
ella.ewens@gmail.com

Reviewer 2 Report
Comments and Suggestions for Authors
The authors, in the later part of the recommendations section suggest it would be helpful to offer appealing, tasty and nutritionally balanced options, menu variety, increasing plant-based options, and removing processed meats throughout the week. Given the push back found in the data, the rationale was not evident for the authors to propose a mandatory national MFM policy should be considered, with the alternative of a meat-less approach considered. Why not propose a predominant plant-based strategy throughout all days while retaining choice for all, including offer animal based options? This may achieve greater gains - on environment, health, and economics - without undermining the principle of freedom of choice.
An additional question that might illuminate the context for these health facilities is around the public policy around food provision. Are their broader government commitments to healthier and more sustainable food policies? Or are these health centres innovating their way in a less-than-supportive environment?
Author Response
Ella Ewens
University of Auckland,
Department of Epidemiology & Biostatistics, School of Population Health, Faculty of Medical and Health Sciences, University of Auckland, Auckland 1023, New Zealand
Ella.ewens@gmail.com
26th October 2023
Subject: Minor Revisions of Manuscript following Peer Review - "Meat Free Mondays in Hospital Cafes in Aotearoa, New Zealand"
Dear Tracy assistant editor, editors and reviewers,
Thank you for your time and consideration in the review of our manuscript. Your care in reviewing the manuscript and interest in the topic is much appreciated.
I am writing to respond to the minor revisions of the manuscript titled "Meat Free Mondays in Hospital Cafes in Aotearoa, New Zealand" from the three peer reviewers, for publication in Nutrients MDPI.
Below are the reviewer comments, and I have outlined how these comments and revisions have been addressed.
|
Reviewer 2 Comment |
Response and actions made |
|
The authors, in the later part of the recommendations section suggest it would be helpful to offer appealing, tasty and nutritionally balanced options, menu variety, increasing plant-based options, and removing processed meats throughout the week. Given the push back found in the data, the rationale was not evident for the authors to propose a mandatory national MFM policy should be considered, with the alternative of a meat-less approach considered. Why not propose a predominant plant-based strategy throughout all days while retaining choice for all, including offer animal based options? This may achieve greater gains - on environment, health, and economics - without undermining the principle of freedom of choice. |
This is a great point, as you point out this could be considered an alternative to MFMs or Meatless Mondays, as stated in the conclusion. A MFM as a single day compared with a week-long plant-rich menu may drive greater awareness, education and strong communications about the environmental and health benefits of reduced meat consumption and therefore change behaviour. The MFM campaign is thought to showcase meat-free meals as the hero, rather than solely making them an option on the daily menu. The present study showed that MFM roll out could have been improved and been more successful with better communication and training of staff. While we agree with your comments, this study only looked at feedback on a MFMs (not a week-long plant-rich menu) so we are unable to advise on whether this would be a more acceptable or effective method to reduce meat consumption. Therefore, we have not amended the text. Another point discussed in US studies by Altema-Johnson and colleagues is the health psychology and behaviour change science involved in the use of Mondays to start a positive habit or change a behaviour. The manuscript has not been amended. |
|
An additional question that might illuminate the context for these health facilities is around the public policy around food provision. Are their broader government commitments to healthier and more sustainable food policies? Or are these health centres innovating their way in a less-than-supportive environment? |
This is a great point. the Eating and Activity guidelines for New Zealand Adults and the National Healthy Food and Drink Policy (NHFDP) in hospitals mention sustainability, although it is a small component and of these guidelines and there are no strong policies or implementation to support it.[15] The wider food environment e.g., the other food outlets in hospital vicinity can be considered broadly unsupportive of MFMs and meat reduction. I have updated the manuscript at L#444 to state the below; For success of MFMs, it is important to consider how supportive the wider food provision policy environment in NZ is of sustainable eating and meat reduction. The wider food environment such as the food outlets in hospital vicinity can be considered broadly unsupportive of MFMs and meat reduction. While the Eating and Activity guidelines for NZ Adults and the NHFDP in hospitals mention sustainability, it is a very small component of these guidelines and there are no strong policies or implementation to support it.[15]
|
Thankyou again for your thorough review of the manuscript and for your helpful and insightful comments.
I appreciate the opportunity to contribute to the body of evidence on sustainable dietary practices.
Sincerely,
Ella Ewens
MPH(Hons), BOptom(Hons), PGDipHsc, PGDipBus
ella.ewens@gmail.com

Reviewer 3 Report
Comments and Suggestions for Authors
The article Meat-Free Mondays in Hospital Cofes in Aoteoroa, New Zealand aims to identify the enables and barriers of MFM policies at three District Health Boards (DHBs) to offer policy recommendations for the future to facilitate implementation, providing original survey data on eating habits and ‘cutting back on meat’ behaviors, attitudes towards MFMs, enablers (e.g., health, environmental, cost, taste), perceived benefits, barriers and impact on future behaviors.
The topic of the manuscript is relevant to the Journal and important, since there are few studies of the effectiveness of MFM Global Campaign.
The methods used are well described and results well structured. However in number of paragraphs there is an implication that the health concerns of the MFM program are more significant that their environmental impact. For instance, L49-L51 the reference [8] is only partially quoted, since the meta-analysis of epidemiological studies stress that “the long-term consumption of increasing amounts of red meat and particularly of processed meat is associated with an increased risk of total mortality, cardiovascular disease, colorectal cancer and type 2 diabetes, in both men and women.” In discussion section L330-L333 the health motivation factor is well identified, however the environmental awareness of the Campaign should find wider place in the Policy Recommendations section L485-495.
The references are appropriate and more that 15 of them are after 2020 year.
Author Response
Ella Ewens
University of Auckland,
Department of Epidemiology & Biostatistics, School of Population Health, Faculty of Medical and Health Sciences, University of Auckland, Auckland 1023, New Zealand
Ella.ewens@gmail.com
26th October 2023
Subject: Minor Revisions of Manuscript following Peer Review - "Meat Free Mondays in Hospital Cafes in Aotearoa, New Zealand"
Dear Tracy assistant editor, editors and reviewers,
Thank you for your time and consideration in the review of our manuscript. Your care in reviewing the manuscript and interest in the topic is much appreciated.
I am writing to respond to the minor revisions of the manuscript titled "Meat Free Mondays in Hospital Cafes in Aotearoa, New Zealand" from the three peer reviewers, for publication in Nutrients MDPI.
Below are the reviewer comments, and I have outlined how these comments and revisions have been addressed.
|
Reviewer 3 Comments |
Response and actions made |
|
The article Meat-Free Mondays in Hospital Cafes in Aoteoroa, New Zealand aims to identify the enables and barriers of MFM policies at three District Health Boards (DHBs) to offer policy recommendations for the future to facilitate implementation, providing original survey data on eating habits and ‘cutting back on meat’ behaviors, attitudes towards MFMs, enablers (e.g., health, environmental, cost, taste), perceived benefits, barriers and impact on future behaviors. The topic of the manuscript is relevant to the Journal and important, since there are few studies of the effectiveness of MFM Global Campaign. The methods used are well described and results well structured. However in number of paragraphs there is an implication that the health concerns of the MFM program are more significant that their environmental impact. For instance, L49-L51 the reference [8] is only partially quoted, since the meta-analysis of epidemiological studies stress that “the long-term consumption of increasing amounts of red meat and particularly of processed meat is associated with an increased risk of total mortality, cardiovascular disease, colorectal cancer and type 2 diabetes, in both men and women.” In discussion section L330-L333 the health motivation factor is well identified, however the environmental awareness of the Campaign should find wider place in the Policy Recommendations section L485-495. |
Thank you for raising this important point. Environmental concerns were thought to be the most important benefit of MFMs across staff interviewed and surveyed in this study,. As you point out, literature shows that consumption of red and processed meat is an important risk factor for non-communicable diseases. In response to your comments I have updated the manuscript at L#49 to the suggested text. The long-term consumption of increasing amounts of red meat and particularly processed meat, is associated with an increased risk of total mortality, cardiovascular disease, cancers such as colorectal cancer and Type 2 diabetes in both women and men. I have also updated the manuscript at L#485 to highlight the importance of the health benefits of MFMs. Consultation should involve key stakeholders, staff, public health dietitians and cultural advisors. Information sharing, training, and evidence-based education to support the environmental and health benefits of the campaign. |
|
The references are appropriate and more that 15 of them are after 2020 year. |
No changes have been made to the references. |
Thankyou again for your thorough review of the manuscript and for your helpful and insightful comments.
I appreciate the opportunity to contribute to the body of evidence on sustainable dietary practices.
Sincerely,
Ella Ewens
MPH(Hons), BOptom(Hons), PGDipHsc, PGDipBus
ella.ewens@gmail.com

Reviewer 4 Report
Comments and Suggestions for Authors
Dear Authors,
Thank you for your manuscript.
You wrote an interesting paper that aimed identify the enablers and barriers of MFM policies at three District Health Boards(DHBs)to offer policy recommendations for the future to facilitate implementation.
The authors have discussed the very relevant and timely topic of the impact of meat consumption on health and the environment. Meatless Mondays (MMs), as described by the authors, is a global campaign to reduce meat consumption that encourages people to give up meat for one day a week.
Some comments and suggestions:
1. in the introduction the authors mention that, most New Zealanders consume below the recommended intake of vegetables, fruits and fiber, it would be good to add some publication and state at what level the current intake of fruits, vegetables and fiber in New Zealand is.
2.It would be good for the online survey for employees to be included in the supplement so that the reader can refer to it.
3. In the methodology, the authors write that they chose three DHB locations at different stages of MFM implementation. They mention that they had previously informally surveyed Northland DHB employees (DHB2), about MFM. However, in the results and discussion section, the authors do not mention the results obtained at DHB2 at all. So it is not understood why the authors even mention this location?
4. In the results section, the "p" values should be in Table 2 and Table 3. It is a mistake to give selected p values only in the text.
5. part of the results in section 3.3 Enablers and Barriers to Cutting Back on Meat it is worthwhile that it is presented in the form of a table, it would be more readable.
6) The presentation of the results is a bit chaotic.
7.The conclusions are too broad it is more of a summary , they should be rewritten.
8. discussion sentence only three of the 20 DHBs were evaluated, it is worth adding this to the research methodology.
Kind regards,
Comments on the Quality of English Language
Minor editing of English language required.
Author Response
Ella Ewens
University of Auckland,
Department of Epidemiology & Biostatistics, School of Population Health, Faculty of Medical and Health Sciences, University of Auckland, Auckland 1023, New Zealand
Ella.ewens@gmail.com
27th October 2023
Subject: Minor Revisions of Manuscript following Peer Review - "Meat Free Mondays in Hospital Cafes in Aotearoa, New Zealand"
Dear Tracy assistant editor, editors and reviewers,
Thank you for your time and consideration in the review of our manuscript. Your care in reviewing the manuscript and interest in the topic is much appreciated.
I am writing to respond to the minor revisions of the manuscript titled "Meat Free Mondays in Hospital Cafes in Aotearoa, New Zealand" from the fourth peer reviewer, for publication in Nutrients MDPI.
Below are the reviewer comments, and I have outlined how these comments and revisions have been addressed.
|
Reviewer 4 Comment |
Response and actions made |
|
1. in the introduction the authors mention that, most New Zealanders consume below the recommended intake of vegetables, fruits and fiber, it would be good to add some publication and state at what level the current intake of fruits, vegetables and fiber in New Zealand is. |
Great idea, I have added a sentence at line #46 to cover this. Please note that NZ is lacking up-to-date food consumption data.
The most recent ) NZ Adult Nutrition Survey (2008/09) indicates that only 32.8% of NZ adults meet the combined recommended vegetable and fruit intake of at least three servings of vegetables and at least two servings of fruit each day.[5] For fibre, NZ adults are consuming less than the daily adequate intake of 30g and 25g respectively, mean intakes for male and female adults were 22.1g and 17.5g.[5] |
|
2.It would be good for the online survey for employees to be included in the supplement so that the reader can refer to it. |
Thanks for the suggestion, I have added this as a supplementary file. I have referred to it in the text at line#97, as per Nutrients guidelines. (Document S1 Meat-Free Mondays Survey).
|
|
3. In the methodology, the authors write that they chose three DHB locations at different stages of MFM implementation. They mention that they had previously informally surveyed Northland DHB employees (DHB2), about MFM. However, in the results and discussion section, the authors do not mention the results obtained at DHB2 at all. So it is not understood why the authors even mention this location? |
The study is made up of interviews and surveys. DHB2 is included in the interviews. This is stated in the results. I have added it in the results section, to improve clarity.
I have updated the following text in the manuscript at L#99 to make it clearer. DHB2 was included in the interviews.
|
|
4. In the results section, the "p" values should be in Table 2 and Table 3. It is a mistake to give selected p values only in the text. |
Thanks for this suggestion. P values have been added as footnotes to Table 2 and Table 3 (now Table 4). |
|
5. part of the results in section 3.3 Enablers and Barriers to Cutting Back on Meat it is worthwhile that it is presented in the form of a table, it would be more readable. |
Thanks for this suggestion, I have added a table (Table 3) which includes the enablers and barriers.
|
|
6) The presentation of the results is a bit chaotic.
|
The results are presented in this order
We think this is the most logical order to present them. For clarity, at the beginning of the results section, I have added a sentence to explain that this is the order that the results will be presented.
Results of the surveys at DHB1 and DHB3 are presented first, followed by the interview results for DHB1, DHB2, and DHB3. |
|
7.The conclusions are too broad it is more of a summary , they should be rewritten.
|
I have edited the conclusion, to make it less broad.
See below which changes highlighted (see also attached document).
This research examined MFM policies at three DHBs in NZ at different stages of MFM implementation and identified enablers and barriers at organisational and individual levels. Awareness of MFMs was high, with strong support from both staff and managers, despite some criticism. Resistance of staff to change behaviour, concerns around special diets and retaining personal choice were identified as key barriers. Over half of the surveyed staff were reducing their meat intake, driven by health and environmental concerns. Qualitative analysis uncovered key themes, including the importance of food quality and appearance for business success and customer retention, communication, education and implementation success, and the overarching significance of hospitals taking the lead in human and planetary health and fostering healthy, sustainable diets. Policy recommendations for successful implementation of MFMs include wide consultation with food service, management, hospital staff, and cultural advisors; providing clear aims and alternatives; considering the wider food environment (including NZ food policy and food outlets in the hospital area); and providing full training and support for the food service, including dietitian support. Communication, education and providing evidence of the benefits of MFMs are crucial factors for a successful MFM policy roll-out. Evaluation of the policy's environmental impact and costs will be valuable in the future as the health system remains a high emitter of greenhouse gas emissions. MFM policies can drive awareness of public health messages about the impact of meat consumption, and despite barriers, comprehensive implementation is likely to make MFMs successful in healthcare settings.
|
|
8. discussion sentence only three of the 20 DHBs were evaluated, it is worth adding this to the research methodology.
|
I have added this to line#91 of the methods section.
Three DHBs were selected (of the total 20 DHBs in NZ) at different stages of contemplation or implementation of MFMs. |
Thank you again for your thorough review of the manuscript and for your helpful and insightful comments.
I appreciate the opportunity to contribute to the body of evidence on sustainable dietary practices.
Sincerely,
Ella Ewens
MPH(Hons), BOptom(Hons), PGDipHsc, PGDipBus
ella.ewens@gmail.com
